# Practical Barriers and Facilitators Experienced by Patients, Pharmacists and Physicians to the Implementation of Pharmacogenomic Screening in Dutch Outpatient Hospital Care—An Explorative Pilot Study

**DOI:** 10.3390/jpm10040293

**Published:** 2020-12-21

**Authors:** Pauline Lanting, Evelien Drenth, Ludolf Boven, Amanda van Hoek, Annemiek Hijlkema, Ellen Poot, Gerben van der Vries, Robert Schoevers, Ernst Horwitz, Reinold Gans, Jos Kosterink, Mirjam Plantinga, Irene van Langen, Adelita Ranchor, Cisca Wijmenga, Lude Franke, Bob Wilffert, Rolf Sijmons

**Affiliations:** 1Department of Genetics, University Medical Center Groningen, University of Groningen, 9713 GZ Groningen, The Netherlands; w.h.drenth@umcg.nl (E.D.); l.g.boven@umcg.nl (L.B.); amandavanhoek@home.nl (A.v.H.); annemiekhijlkema@hotmail.nl (A.H.); ellenpoot@hotmail.com (E.P.); g.b.van.der.vries@umcg.nl (G.v.d.V.); m.plantinga@umcg.nl (M.P.); i.m.van.langen@umcg.nl (I.v.L.); c.wijmenga@rug.nl (C.W.); l.h.franke@umcg.nl (L.F.); r.h.sijmons@umcg.nl (R.S.); 2Department of Psychiatry, University Medical Center Groningen, University of Groningen, 9713 GZ Groningen, The Netherlands; r.a.schoevers@umcg.nl (R.S.); Ernst.Horwitz@ggzfriesland.nl (E.H.); 3Department of Internal Medicine, University Medical Center Groningen, University of Groningen, 9713 GZ Groningen, The Netherlands; r.o.b.gans@umcg.nl; 4Department of Clinical Pharmacy and Pharmacology, University Medical Center Groningen, University of Groningen, 9713 GZ Groningen, The Netherlands; j.g.w.kosterink@umcg.nl (J.K.); b.wilffert@rug.nl (B.W.); 5Unit of PharmacoTherapy, Epidemiology & Economics, Groningen Research Institute of Pharmacy, University of Groningen, 9713 AV Groningen, The Netherlands; 6Department of Health Psychology, University Medical Center Groningen, University of Groningen, 9713 GZ Groningen, The Netherlands; a.v.ranchor@umcg.nl

**Keywords:** pharmacogenetics, pharmacogenomics, implementation, screening, pre-emptive, personalized medicine, precision medicine

## Abstract

Pharmacogenomics (PGx) can provide optimized treatment to individual patients while potentially reducing healthcare costs. However, widespread implementation remains absent. We performed a pilot study of PGx screening in Dutch outpatient hospital care to identify the barriers and facilitators to implementation experienced by patients (*n* = 165), pharmacists (*n* = 58) and physicians (*n* = 21). Our results indeed suggest that the current practical experience of healthcare practitioners with PGx is limited, that proper education is necessary, that patients want to know the exact implications of the results, that healthcare practitioners heavily rely on their computer systems, that healthcare practitioners encounter practical problems in the systems used, and a new barrier was identified, namely that there is an unclear allocation of responsibilities between healthcare practitioners about who should discuss PGx with patients and apply PGx results in healthcare. We observed a positive attitude toward PGx among all the stakeholders in our study, and among patients, this was independent of the occurrence of drug-gene interactions during their treatment. Facilitators included the availability of and adherence to Dutch Pharmacogenetics Working Group guidelines. While clinical decision support (CDS) is available and valued in our medical center, the lack of availability of CDS may be an important barrier within Dutch healthcare in general.

## 1. Introduction

Pharmacogenomics (PGx) studies the interplay between variation in the human genome and drug response. Knowledge about PGx can help predict which medication will be most effective and safe in individual patients while potentially reducing healthcare costs [1,2]. Different approaches to apply PGx knowledge in patient care exist. On one hand, PGx testing can be performed in a reactive manner to find an explanation for a low therapeutic response or the occurrence of adverse drug reactions (ADRs). On the other hand, ideally, an individual’s PGx profile is already known before drug prescription, so treatment can be tailored to the individual’s genome without awaiting potential treatment failure or the occurrence of ADRs. This approach is known as pre-emptive PGx testing or PGx screening.

The potential benefits of introducing PGx screening into a routine healthcare setting include reduced hospitalizations and cost and improved safety, adherence and efficacy [3]. Dutch national guidelines on the practical application of PGx for drug prescription of 95 drugs, developed by the Dutch Pharmacogenetics Working Group (DPWG), are available through the Dutch drug database, referred to as the G-standard [4,5]. Based on these DPWG guidelines, it is estimated that an alternative dosage or drug would be recommended for 1 in 20 drug prescriptions in primary care if PGx screening became the standard-of-care in the Netherlands [6]. Nevertheless, PGx is rarely applied in current clinical practice [2,7].

A number of barriers to PGx implementation have been identified so far. These include unclear procedures, insufficient evidence, inefficient infrastructure, lack of a standardized format for reporting results, lack of ICT support tools, and lack of knowledge, training and experience among healthcare practitioners. Reported facilitators include recognition of clinical utility, pharmacist’s feelings of responsibility for delivering PGx to patients, and the availability of professional guidelines for interpreting test results [1,8,9,10,11,12,13]. To the best of our knowledge, no study has identified barriers and facilitators from the perspective of all the relevant stakeholders in an actual implementation setting. Therefore, we carried out an explorative pilot study to identify such barriers and facilitators while offering PGx screening in two outpatient clinics of the University Medical Center of Groningen (UMCG) in the Netherlands.

## 2. Materials and Methods

This study was designed as an explorative pilot study with mixed methods. The study timeline is shown in Figure 1A, and an overview of the study design in Figure 1B. Additional background information is provided in Appendix A.

### 2.1. Recruitment of Participants

The outpatient clinics of Internal Medicine and Psychiatry and the hospital pharmacy of the UMCG were approached to participate in this study. Information about the study’s aim was provided during an introductory meeting with each department. Physicians who took part recruited participants from their own patients on a first-come-first-served basis until the study test capacity of 165 PGx individuals was reached. Inclusion criteria were: 18 years or older, cognitively competent, able to read and speak Dutch, and able to provide a blood sample. Eligible patients received printed information about the project goal, procedures for testing, reporting of results and links to resources with additional information (project website and animated video). Copies of these materials (in Dutch) are available upon request.

Community pharmacists listed in the patient’s electronic health record (EHR) were invited to fill out questionnaires by mail simultaneously with the reporting of PGx screening results (T1). UMCG physicians at the two clinics and hospital pharmacists involved in patient care were invited to fill out study questionnaires by email at the end of follow-up (T2, Figure 1A). See Appendix A for additional details.

### 2.2. Genotyping and Reporting of PGx Screening Results

After providing written informed consent, patients underwent genotyping with a custom panel of 14 genes (Appendix A). Next, patients received a letter with their PGx screening results and an explanation in layman’s terms (see Appendix A). Copies were also stored in their hospital EHR and sent to their community pharmacist and general practitioner (GP) (Figure 1B). See Appendix A for additional details.

Custom clinical decision support (CDS) software developed prior to the study was used to provide hospital prescribers with relevant DPWG recommendations in real time during drug prescription (Figure 1). See Appendix A for additional details.

### 2.3. Data Collection

PGx screening results, predicted drug-gene interactions (DGIs), and CDS use, including user comments and actions that were taken based on recommendations, were stored in the study database. Relevant medical information, including patient drug use during the follow-up period November 2017–November 2018, was manually extracted from EHRs (see Appendix A for additional details). Follow-up started from the time the results were reported and therefore varied between patients, up to a maximum of a year (Figure 1A). We conducted five questionnaires to evaluate the experiences of patients, physicians, and pharmacists via open- and closed-ended questions at the time of result reporting (T1) and after follow-up (T2, Figure 1A). The survey study was designed by a multidisciplinary team using input from an explorative qualitative interview and focus group study with 13 prescribers from the participating outpatient clinics, 13 patients and 7 pharmacists (see Appendix A). The questionnaires included items on various themes: sociodemographics, knowledge and education about PGx, attitude towards PGx screening, application of PGx, provision of information about PGx, and result reporting (Appendix A). The attitude questions originate from the theory of planned behavior framework [14]. All other questions were self-constructed. The two patient questionnaires were sent out on paper, with the option to respond digitally, at the time of results reporting (T1) and after follow-up (T2). If necessary, patients were reminded by mail and again by telephone to respond. Community pharmacists were invited to respond to the survey on paper, with the option to respond digitally at the time results were reported (T1). The outpatient clinic physicians and hospital pharmacists received an invitation for a digital survey by email after follow-up (T2, Figure 1A). Digital survey responses were collected using the routine outcome monitoring application RoQua [15]. Responses on paper were registered in RoQua by the researchers.

### 2.4. Data Analysis

CDS searches and survey responses to open-ended questions were independently categorized by two researchers (AvH, AMAH), and discrepancies were resolved by a third independent researcher (PL). All data collected was pseudonymized and analyzed per theme using R [16]. For survey responses, the Shapiro–Wilk test was used to assess normality. Subsequent subgroup comparisons were performed using a *t*-test or Wilcoxon test. Cronbach’s alpha was used to assess the internal consistency of survey questions.

### 2.5. Ethical Approval

This study was approved by the Medical Ethics Review Board of the UMCG (reference: 2017.266).

## 3. Results

### 3.1. Participants

This study included 165 patients, 21 physicians, 13 hospital pharmacists, and 48 community pharmacists (Figure 1B) and explored various themes around practical barriers and facilitators. Response rates to the patient questionnaires were 84% (*n* = 138, T1) and 74% (*n* = 122, T2). Response rates to the healthcare practitioner questionnaires were 19% (physicians, T2), 28% (hospital pharmacists, T2), and 77% (community pharmacists, T1). Response rates per survey item varied since not all respondents have answered all items. Median patient follow-up was 244 days (range: 117–365). See Appendix A for the full demographics of study participants.

### 3.2. Screening Results, Drug Use and DGIs

Out of the study population, 158 patients (96% of the study population) carried at least one actionable PGx haplotype or predicted PGx phenotype (Appendix A lists frequencies of PGx haplotypes and predicted PGx phenotypes). During follow-up, 60 patients received drug treatment (36%). Following DPWG guidelines, DGIs were observed in 21 patients (13%): 18 with one DGI, one with two DGIs and two with three DGIs. Actionable DGIs were observed in 20 patients (12%): 18 with one actionable DGI and two-with-two actionable DGIs. In total, 120 unique drugs were used during follow-up, including 18 with a known DGI (15%), of which 15 were actionable in the study population. During follow-up, patients used two drugs (range: 0–13 drugs) on average, and 27 patients (23% of T2 respondents) reported being prescribed at least one new drug. Patients reported that prescriptions originated from their GP’s office (83% of T2 respondents) or hospital physician (17% of T2 respondents). See Appendix A for survey results on the review of patient drug use in response to PGx screening results.

### 3.3. CDS Searches and Output during Follow-Up

During follow-up, CDS was used to consult the DPWG guidelines 59 times for 20 patients. A CDS search was performed for eight patients who received drug treatment, and four had DGIs. CDS searches were categorized into six subgroups using treatment information from the EHR: prescribing situation (5%), cascade (search in response to the previous search) (2%), potential future treatment (29%), current treatment (47%), past treatment (5%), and other (12%). Of the CDS searches, 27 (45.8%) yielded recommendations requiring an action by the prescriber, 14 (23.7%) did not require an action, 10 (16.9%) found no available recommendations (e.g., in case of normal metabolizers), and 8 (13.6%) had inconclusive test results. Of the actionable recommendations, 12 (44%) advised adhering to an adjusted maximum (daily) dose or prescribing an alternative, 5 (19%) advised prescribing an alternative, 5 (19%) advised lowering the dose and monitoring plasma concentrations, 2 (7%) advised adjusting the dose based on the effect observed, 2 (7%) advised lowering the maintenance dose, and 1 (4%) advised increasing the dose. Details of the DGIs involved and an evaluation of DPWG guidelines are presented in Appendix A.

### 3.4. Prior Experience of Healthcare Practitioners with PGx

Twenty-one community pharmacists (44% of respondents), one hospital pharmacist (8%), and five physicians (24%) reported that this study was their first experience with PGx test results. One in eight community pharmacists, six hospital pharmacists (46%) and half of the physicians (52%) reported having taken the initiative to conduct PGx testing at least once in the past. These results highlight that the current practical experience is limited.

### 3.5. Knowledge and Education of Healthcare Practitioners

In all professions, half the healthcare practitioners participating in this study reported having received postgraduate education about PGx. The self-graded knowledge level was significantly higher in these subgroups (Table 1). Pharmacists reported a need for further education, both for themselves (*n* = 47, 77%) and for pharmacy staff (*n* = 52, 87%), whereas physicians did not report this need.

### 3.6. Patient Attitudes towards PGx Screening after Follow-Up (T2)

Most patients reported that genetic testing in general (*n* = 89, 77% of T2 respondents) or PGx testing (*n* = 102, 88%) did not frighten them. Knowing their PGx profile was considered comforting (*n* = 106, 89%) and useful (*n* = 111, 92%), and patients thought that it has added value when their pharmacotherapy is adjusted using PGx (*n* = 107, 91%). No significant difference was found in the attitude of patients with or without observed DGIs.

### 3.7. Healthcare Practitioner Attitudes towards PGx Screening

Nearly all healthcare practitioners were positive about the usefulness of PGx information for their patients (useful to have *n* = 69, 84% of respondents; would like to use more in daily practice: *n* = 72, 88%; added value: *n* = 71, 87%). However, nine community pharmacists (19%), two hospital pharmacists (15%) and four physicians (20%) did not feel ready to apply PGx information in daily practice.

### 3.8. Practical Application of PGx

Community pharmacists graded their expected application level (T1), whereas hospital pharmacists and physicians graded their perceived application level (T2). The self-graded application level is significantly higher in the education subgroups for both community and hospital pharmacists, but not for physicians (Table 1). Prominent arguments provided to explain higher self-graded application levels were that application of PGx was possible with the use of the pharmacy or hospital computer system (*n* = 12) and that healthcare practitioners had come across PGx more often (during education or in practice) (*n* = 8). Notable arguments to explain lower self-graded application levels were that healthcare practitioners perceived insufficient knowledge themselves (*n* = 8) and reported practical barriers present within computer systems, for example, that not all PGx results could be registered (*n* = 5). In summary, healthcare practitioners relied heavily on their computer system for the application of PGx, perceived a need for education on PGx application, and experienced practical barriers within computer systems that hindered PGx application. Appendix A describes an event that occurred during follow-up that illustrates the importance of educating and informing all healthcare practitioners involved in the practical application of PGx.

### 3.9. Patients’ Needs for Information about Their PGx Screening Results

After receiving the PGx screening results, 15 patients (11% of T1 respondents) reported still having questions with respect to these results, most often wanting to know the exact implications, e.g., the level of dose adjustment or suitable alternative drugs (*n* = 6). Patients generally consulted their treating physician in the hospital during follow-up to gain additional information. After follow-up, the number of patients having questions about their PGx screening results has increased to 23 (19% of T2 respondents). They still primarily wanted to know the implications of the results for them (*n* = 7). Thirty-six patients (30% of T2 respondents) reported that improvements could be made in the information provided, most importantly in explaining the exact implications of the results for them (*n* = 9), providing better explanation in general (*n* = 7), and better educating healthcare practitioners (*n* = 4).

A detailed evaluation of the PGx result letter is presented in Appendix A. In summary, some patients wished to receive more and different information than provided in this study.

### 3.10. Discussing PGx Screening Results with Patients: Patient Surveys

After receiving the PGx screening results, 47 patients (35% of T1 respondents) believed a healthcare practitioner should always discuss these results with them, 29 (21%) only if patients express the need, and 33 (24%) only if the results have consequences. Twenty-six (19%) thought the results should not be discussed with them at all. According to patients, the preferred healthcare practitioners to discuss PGx screening results are the treating physician in the hospital (*n* = 80, 44%), GP (*n* = 47, 26%), clinical geneticist (*n* = 30, 16%), or pharmacist (*n* = 22, 12%).

After receiving the PGx screening results, 101 patients (74% of T1 respondents) planned to discuss them with their treating physician, with 44 patients (37% of T2 respondents) reporting having done so after follow-up in a regular appointment and 6 (5%) reporting having done so in a separate appointment. In total, 101 conversations about PGx screening results between patients and healthcare practitioners were scored by patients (46% physician, 21% community pharmacist, 21% GP, 8% physician from another hospital, 2% home nurse, 2% thrombosis care, and 1% nursing home). Seventy-one percent of these conversations were scored as “(very) good”. In one case, the conversation was scored as “good”, but the patient reported that the healthcare practitioner did not (fully) understand the results. Thirteen percent of conversations were scored as “(very) bad”. In two cases, the conversation as such was scored as “(very) bad” even though, on a positive note, the healthcare practitioner had started using the PGx results (Figure 2).

### 3.11. Discussing PGx Screening Results with Patients: Healthcare Practitioner Surveys

Sixteen community pharmacists (36% of respondents), eight hospital pharmacists (62%) and 13 physicians (62%) believed that PGx screening results should always be discussed with patients by a healthcare practitioner, with eight (18%), two (15%) and five (24%), respectively, believing it should only be done if a patient expresses the need and 19 (42%), three (23%) and three (14%), respectively, only if the results have consequences. Two community pharmacists (4%) did not believe results should be discussed with patients at all. Community pharmacists primarily placed the responsibility for discussing PGx screening results with patients in the hands of the treating physician in the hospital (*n* = 26, 38%) or pharmacist (*n* = 21, 31%), and to a lesser extent with the clinical geneticist (*n* = 13, 19%). Hospital pharmacists also primarily placed this responsibility in the hands of the treating physician in the hospital (*n* = 11, 39%) or pharmacist (*n* = 8, 29%), and to a lesser extent with the GP (*n* = 4, 14%) or clinical geneticist (*n* = 4, 14%). Physicians primarily indicated that they, as treating physicians in the hospital, should discuss PGx screening results with patients (*n* = 19, 59%), followed by the pharmacist (*n* = 7, 22%) and the GP (*n* = 3, 9%).

Community pharmacists were asked what they planned to do with the PGx screening results they had received (T1). All plans reported for PGx screening results are shown in Figure 3. Although four community pharmacists reported that PGx screening results should always be discussed with the patient by a healthcare practitioner, preferably the pharmacist, none of these four pharmacists reported that they themselves intended to discuss the results with their patients.

Five out of six hospital pharmacists, and all eight physicians who discussed PGx screening results with patients and/or other healthcare practitioners felt they had sufficient knowledge to do so. None of them reported questions about PGx that they were unable to answer.

### 3.12. Responsibility for Application of PGx Screening Results in Patient Care

Healthcare practitioners were also asked about whom they regarded as having the final responsibility for the application of PGx screening results in patient care. The results are presented in Table 2 and show that the majority of physicians reported that this responsibility lies with the prescriber. Hospital pharmacists largely agreed with this, although a notable group also reported the pharmacist as responsible. Community pharmacists were more divided and specifically indicated that there is a shared responsibility. In summary, the allocation of responsibility for the application of PGx screening results in patient care is currently unclear.

### 3.13. Identified Practical Barriers and Facilitators

An overview of the identified practical barriers and facilitators within the various themes discussed above, as perceived by healthcare practitioners and patients, is presented in Table 3.

## 4. Discussion

This study identified practical barriers and facilitators within various themes, as perceived by healthcare practitioners and patients, to the use of PGx screening results and associated DPWG recommendations in a Dutch outpatient hospital care setting (Table 3). As some of the survey questions dealt with the actual outcome of PGx testing, we discuss these first.

### 4.1. Frequencies of PGx Variants and DGIs

We confirmed that actionable PGx variants are present in the majority of the patient population of outpatient clinics in frequencies comparable to those reported in the literature (Appendix A). Since the majority of new prescriptions during follow-up originated from the GP, and drugs prescribed by GPs were not considered in our study, the number of DGIs we report is likely an underestimation. It is important that the number of DGIs is determined in more detail for a variety of patient populations in order to assess the value of PGx for individual patients.

CDS searches were performed in only four patients with a DGI, but recommendations were shown for more patients. This is explained by the fact that an alternative drug without a DGI was prescribed following the recommendation shown or because drugs were not prescribed directly following the search. The latter is illustrated by the search types we could distinguish. Some searches concerned past or future treatment, and prescribers also checked drugs they did not want to prescribe at that moment, for example, commonly used treatment alternatives or drugs that were suggested in a recommendation. Furthermore, it is likely that prescribers started to remember the recommendations for DGIs they had encountered previously and did not perform a CDS search every time. The number of CDS searches reported is therefore likely to be an underestimation of the actual number of times prescribers dealt with PGx results.

From the actionable recommendations evaluated, we conclude that DPWG guidelines are generally well adhered to, although the practical application can transcend guideline recommendations, and application is thus not always straightforward.

### 4.2. Practical Barriers and Facilitators

In agreement with the literature, our results show that current practical experience with PGx is limited, even though DPWG guidelines have been available nationwide since 2006 [2,4,7]. A lack of knowledge and training among healthcare practitioners has previously been reported as a barrier to PGx implementation [1,8,9,10,12,13]. The community and hospital pharmacists in our study reported wanting more education about PGx for themselves and pharmacy staff. Physicians in our study did not report this, which does not directly imply that they have enough knowledge or skills, given that some also reported not feeling ready to apply PGx in daily practice. While physicians themselves perceived the general introduction and presentation of DPWG guideline recommendations provided in this study as sufficient, some patients wanted physicians to be better informed. According to these patients, some physicians were unable to provide sufficient explanation or did not fully understand the results. Our findings suggest that postgraduate education could increase the ability of healthcare practitioners to apply PGx in practice. Due to the explorative nature of our study, we can only speculate that the currently available training may not correspond well with practical needs (specifically on the topic of communication with patients), that training may not be optimized for physicians, that physicians may be unaware of their lack of knowledge and skills, or that physicians may have a lower demand for in-depth knowledge about PGx in general compared to pharmacists. Further research is needed to investigate the details underlying this barrier.

Literature reports that recognition of the clinical utility of PGx is a facilitator for implementation and that disbelief is a barrier [8,9,10,12,13]. In our study, patients were positive about PGx, including its expected clinical utility, regardless of the occurrence of DGIs during their treatment, whereas healthcare practitioners were generally positive about the clinical utility, although some did not feel ready to apply it in daily practice. These results should, however, be interpreted with caution because patients and healthcare practitioners who recognized the clinical utility were more likely to participate in this study and our study size was limited. In addition, patients and physicians were recruited from only two outpatient clinics, Psychiatry and Internal Medicine, and this may have influenced the outcome, for example, because practical use of reactive PGx testing is relatively common in psychiatry compared to other medical fields.

In our study, PGx screening results were reported directly to patients by mail without the presence of a healthcare practitioner. In the absence of a standardized reporting format for PGx testing results, which has previously been reported as a barrier [9], we drafted a patient result letter with a brief explanation of the results in laymen’s terms and suggested actions, e.g., that the patient discusses their results with their current healthcare practitioners and share results with any new ones. Considering that pharmacotherapy is often a complex balance between treatment options, effectiveness, (risk of) ADRs, co-morbidities, and co-medication, it is our view that communicating the implications should be up to the individual healthcare practitioner and should be tailored to the individual patient at the time it is relevant. Patients should only have to know when to share the PGx screening results with their healthcare practitioner, e.g., in those cases where that information is not routinely included in their EHR. While the patient result letter was developed based on feedback from patients in focus groups prior to the study, our results indicate that some patients wanted to receive more and different information than provided. Most importantly, patients repeatedly reported wanting to know the exact implication of the PGx screening results for them, e.g., the level of dose adjustment or suitable alternative drugs. However, not all patients desired this depth of information, implying that one format for reporting PGx results to all patients would not suffice. An electronic personal health environment could present information to patients about their PGx screening results while containing multiple layers of information that enable them to receive the depth of information they desire, while also providing a standardized reporting format for PGx results and a way for patients to easily share their results with their healthcare practitioners.

A new barrier emerged from our study: the unclear allocation of responsibilities among healthcare practitioners. The majority of patients reported that PGx screening results should be discussed with them by a healthcare practitioner but had differing preferences for which healthcare practitioner should be responsible. We also found that healthcare practitioners themselves perceived they had a shared, and therefore still unclear, responsibility for discussing PGx screening results with patients. It was also unclear to both patients and healthcare practitioners at what point in the treatment process PGx screening results and their implications should be discussed, if ever. It is also unclear which healthcare practitioner is ultimately responsible for the application of PGx screening results in different patient care situations. Furthermore, a group of patients reported their current drugs were not reviewed by a healthcare practitioner even though they desired this (data presented in Appendix A). Although some patient’s drugs may have been reviewed without their knowledge, these results underline the importance of clear communication with patients and expectation management. In addition, we should be aware of the risk of suboptimal pharmacotherapy in situations where patients are unassertive or have a more “wait-and-see” attitude because it is unclear which healthcare practitioner is responsible for discussing and applying PGx in practice. In our opinion, it should never be the patient’s responsibility to make sure PGx screening results are discussed and/or applied. Overall, this newly identified barrier needs to be addressed to facilitate the responsible implementation of PGx screening. However, this may not be easily done nationally or internationally, as the interactions between healthcare practitioners can be highly variable between countries, regions, and even healthcare organizations or healthcare practitioners. As we identified this barrier in our limited local setting, additional research is needed to identify whether an unclear allocation of responsibilities is also a national/international barrier.

For logistical reasons, CDS software was only available as a separate tool outside the EHR in which the drugs are prescribed during our study, which presented a barrier for physicians to consider PGx screening results during prescription. This approach was taken because the availability of CDS software was deemed crucial in our pre-pilot study (see Appendix A), which is supported by the literature [7,12,13]. In response to our explorative pilot study, PGx-based medication surveillance has now been incorporated into our hospital EHR (since July 2020) in order to facilitate the application of DPWG guidelines for every patient, both those admitted and those treated in outpatient clinics. The availability of CDS within our EHR is an important and crucial step towards the use of PGx-based medication surveillance in routine healthcare. However, not all computer systems used by healthcare practitioners outside of our hospital can handle (all) PGx screening results. Since healthcare practitioners rely heavily on their computer system for insight into DPWG guidelines during drug prescription and medication surveillance, the lack of availability of CDS may be an important barrier within Dutch healthcare in general.

In the Netherlands, PGx testing is currently only reimbursed by the insurer to investigate the cause of an ADR or as part of an optional reimbursement package. In anticipation of resolving this financial barrier to the broad implementation of PGx testing and screening, we provided physicians with the opportunity to perform PGx screening for their patients free-of-charge and with minimal selection criteria. This study did not address which patients should be screened and at what time point in their treatment; the costs of PGx screening would be best justified. Further research, including health technology assessment, should inform policy decision-making on these aspects.

To conclude, our exploratory pilot study confirmed known practical barriers and facilitators and suggested a new barrier to the implementation of PGx screening, namely an unclear allocation of responsibilities among healthcare practitioners. With this knowledge, we have more insight into which facilitators can be leveraged and which barriers need to be overcome to successfully implement PGx screening in Dutch outpatient hospital care. This study also provides a foundation for more detailed novel research that will hopefully further aid PGx implementation and contribute to unlocking the full potential of genome-guided drug prescription to enable personalized medication schemes with optimized treatment tolerance and response.

## Figures and Tables

**Figure 1 jpm-10-00293-f001:**
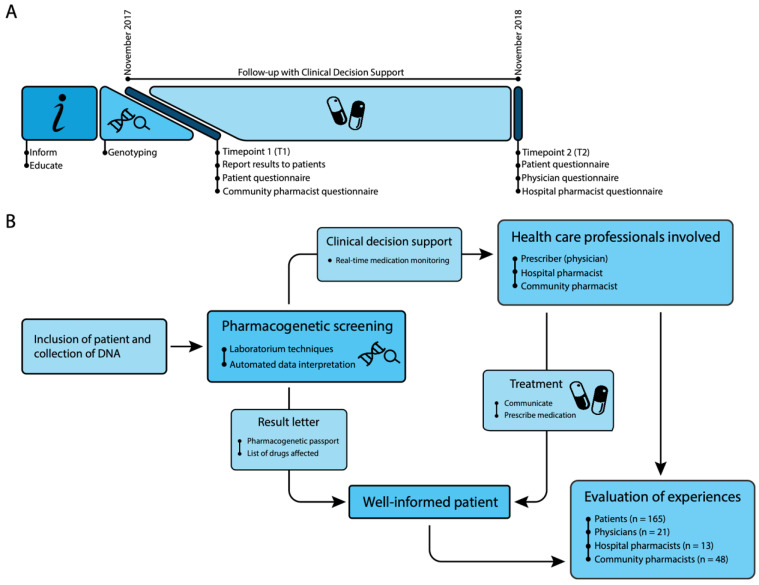
Study timeline and design (**A**). Study timeline (**B**). Study design.

**Figure 2 jpm-10-00293-f002:**
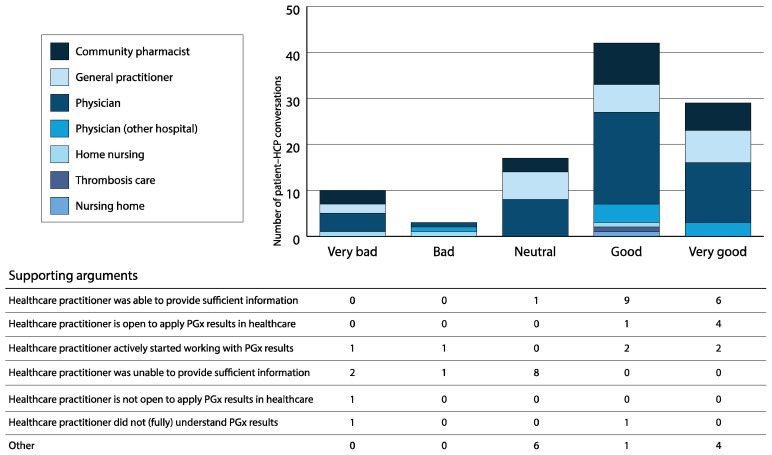
Conversation scores for the discussion of pharmacogenomic test results with healthcare practitioners. The number of conversations between patients and different healthcare practitioners, the score patients gave to those conversations, and the supporting arguments for the score given.

**Figure 3 jpm-10-00293-f003:**
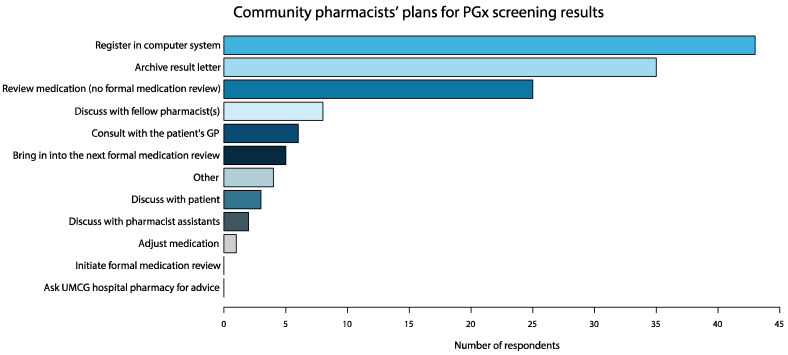
The steps which community pharmacists reported they would take after having received pharmacogenomic screening results.

**Table 1 jpm-10-00293-t001:** Self-graded knowledge and application level of healthcare practitioners.

Healthcare Practitioner		Self-Graded Knowledge Level		Self-Graded Application Level	
Community pharmacists	with postgraduate education	6.5 (4–8)	*p* = 0.011	7 (5–10)	*p* = 0.005
	without postgraduate education	6 (2–7)		5 (2–8)	
Hospital pharmacists	with postgraduate education	7.7 (7–9)	*p* = 0.01	7.5 (7–9)	*p* = 0.016
	without postgraduate education	6.3 (5–7)		6 (2–7)	
Physicians	with postgraduate education	7 (1–9)	*p* = 0.002	7 (6–8)	*p* = 0.203
	without postgraduate education	4 (6–8)		6 (3–9)	

**Table 2 jpm-10-00293-t002:** Final responsibility for the application of pharmacogenomic screening results in patient care.

Responsible Person	Community Pharmacists	Hospital Pharmacists	Physicians
Pharmacist	18 (39%)	5 (38%)	2 (9.5%)
Prescriber	10 (22%)	8 (62%)	16 (76%)
Clinical geneticist	7 (15%)	-	2 (9.5%)
General practitioner	-	-	-
Other		-	
Shared responsibility in general	5 (11%)		
Pharmacist and prescriber are jointly responsible	4 (9%)		
Pharmacist, providing sufficient information transfer	1 (2%)		
Depending on drugs prescribed	-		1 (5%)
Other	1 (2%)		

**Table 3 jpm-10-00293-t003:** Barriers and facilitators to pharmacogenomic screening implementation.

	Perceived by Stakeholder			
	Patient	Community Pharmacist	Hospital Pharmacist	Physician
**Barriers**				
Practical experience is limited	No	Yes	Yes	Yes
Need for further postgraduate education	No	Yes	Yes	No
Rely on computer systems for application	No	Yes	Yes	Yes
Need for education about PGx application	No	Yes	Yes	Yes
Practical barriers within computer systems	No	Yes	Yes	Yes
Lack of information, specifically about exact implications of PGx screening results	Yes	No	No	No
Unclear allocation of responsibilities among healthcare practitioners	Yes	Yes	Yes	Yes
**Facilitators**				
Positive attitude towards PGx	Yes	Yes	Yes	Yes
DPWG guidelines are generally well adhered to	No	No	No	Yes

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
