# Peer review of "Practical Barriers and Facilitators Experienced by Patients, Pharmacists and Physicians to the Implementation of Pharmacogenomic Screening in Dutch Outpatient Hospital Care—An Explorative Pilot Study"

_jpm, 2020, doi:10.3390/jpm10040293_

Round 1

Reviewer 1 Report

The manuscript of P. Lanting et al is well written in English and is certainly original and innovative in its contents. The only suggestion I want to make is to add a major focus on pharmacogenomic screening in the introduction. This would make it easier for the non-expert reader to better understand your work.

Author Response

We thank you and the reviewers for the critical review of our manuscript. We have revised the manuscript according to the comments and suggestions made and discuss these in detail below:

The manuscript of P. Lanting et al is well written in English and is certainly original and innovative in its contents. 

We thank the reviewer for the shared enthusiasm for our paper’s topic and we provide a response to the suggestion below.

Q1 The only suggestion I want to make is to add a major focus on pharmacogenomic screening in the introduction. This would make it easier for the non-expert reader to better understand your work.

A1 We agree that the concept of PGx could be introduced more in detail to non-expert readers and have revised the introduction as follows (line 48-64): 

‘Pharmacogenomics (PGx) studies the interplay between variation in the human genome and drug response. Knowledge about PGx can help predict which medication will be most effective and safe in individual patients while potentially reducing healthcare costs.[1][2] Different approaches to apply PGx knowledge in patient care exist. On the one hand, PGx testing can be performed in a reactive manner, to find an explanation for low therapeutic response or the occurrence of adverse drug reactions (ADRs). On the other hand, ideally, an individual’s PGx profile is already known before drug prescription so treatment can be tailored to the individual’s genome without awaiting potential treatment failure or the occurrence of ADRs. This approach is known as pre-emptive PGx testing or PGx screening. 

Potential benefits of introducing PGx screening into a routine healthcare setting include reduced hospitalizations and cost, and improved safety, adherence and efficacy.[3] Dutch national guidelines on practical application of PGx for drug prescription of 95 drugs, developed by the Dutch Pharmacogenetics Working Group (DPWG), are available through the Dutch drug database, referred to as the G-standard.[4][5] Based on these DPWG guidelines, it is estimated that an alternative dosage or drug would be recommended for 1 in 20 drug prescriptions in primary care if PGx screening became the standard-of-care in the Netherlands.[6] Nevertheless, PGx is rarely applied in current clinical practice.[2][7]

Reviewer 2 Report

in your introduction you identify your study as an explorative study to identify barriers in various stakeholders. However, although this study was not intended as an epidemiological study to the frequencies of PGx variants and their implications eg dose adjustments, they take quite a prominent place in the results and discussion which takes away the attention of the real subject of your study

Secondly, it is not easy to find how you hoped to identify new(?) barriers and in which group. Table S2 mentions the questions but I fail to understand what the answers mean (eg fgen_apoth_16)

In the results there is a mixture of identified barriers to apply PGx and problems with implementation of the results eg patients attitude towards PGx and need for further information about the results. Last point is very interesting but does it refrain a patient from taking the PGx? 

a table with abbreviations would be helpful

supportive material in Dutch is not relevant for most readers

Author Response

We thank the reviewer for the constructive feedback provided to us. We provide a point-by-point response to the comments below.

Q1 In your introduction you identify your study as an explorative study to identify barriers in various stakeholders. However, although this study was not intended as an epidemiological study to the frequencies of PGx variants and their implications eg dose adjustments, they take quite a prominent place in the results and discussion which takes away the attention of the real subject of your study

A1 We acknowledge that the screening results and their implication take a prominent place in both the results and the discussion, and can draw away the attention from the main focus of our study. We have therefore sought a new balance, thinking it is important to provide enough context for the core topic findings. For example, when no PGx variant would be present in our patient population or no relevant drugs would be prescribed, the results would tell a much more hypothetical story as opposed to the situation we describe. We have therefore condensed the relevant paragraphs in both the results and discussion as much as possible without losing the necessary context. We also removed figure 2. The changes made and resulting condensed paragraphs are listed below. 

Results section, paragraph 3.3 CDS searches and output during follow-up”

  • Removed:
    ‘CDS searches were performed by prescribers from participating outpatient clinics, who were explicitly instructed to use CDS to consult DPWG guidelines, and other prescribers in the hospital, who were either informed about the PGx screening results by the patient or encountered the results in the EHR.
  • Removed:
    Figure 2 and references to this figure.
  • Resulting condensed paragraph (line 184-196):
    ‘During follow-up, CDS was used to consult the DPWG guidelines 59 times for 20 patients. A CDS search was performed for eight patients who received drug treatment, and four had DGIs. CDS searches were categorized into six subgroups using treatment information from the EHR: prescribing situation (5%), cascade (search in response to previous search) (2%), potential future treatment (29%), current treatment (47%), past treatment (5%), and other (12%). Of the CDS searches, 27 (45.8%) yielded recommendations requiring an action by the prescriber, 14 (23.7%) did not require an action, 10 (16.9%) found no available recommendations (e.g. in case of normal metabolizers), and 8 (13.6%) had inconclusive test results. Of the actionable recommendations, 12 (44%) advised adhering to an adjusted maximum (daily) dose or prescribing an alternative, 5 (19%) advised prescribing an alternative, 5 (19%) advised lowering the dose and monitoring plasma concentrations, 2 (7%) advised adjusting the dose based on the effect observed, 2 (7%) advised lowering the maintenance dose and 1 (4%) advised increasing the dose. Details of the DGIs involved and an evaluation of DPWG guidelines are presented in Supplementary Results sections 2 and 3.”

Discussion, paragraph 4.1 Frequencies of PGx variants and DGIs:

  • Removed:
    ‘To the best of our knowledge, this is the first study to report the number of DGIs after PGx screening where the screening was not initiated by the prescription of a drug with a known DGI. The median number of drugs used by participating patients during follow-up was two, whereas the median number of drug dispensed to a patient in the same Northern Netherlands regions, as registered in the IADB.nl database, was three.[17] We also only analyzed drugs recorded in the EHR.’
  • Removed:
    ‘Although a comparison to standard-of-care is not possible based on available data, and we cannot discriminate between the different reasons for the changes made after drug review during this study, it is likely that DPWG recommendations altered prescription choices.’
  • Resulting condensed paragraph (line 410-450):
    ‘We confirmed that actionable PGx variants are present in the majority of the patient population of outpatient clinics in frequencies comparable to those reported in literature (Table S4). Since the majority of new prescriptions during follow-up originated from the GP, and drugs prescribed by GPs were not considered in our study, the number of DGIs we report is likely an underestimation. It is important that the number of DGIs is determined in more detail for a variety of patient populations in order to assess the value of PGx for individual patients.

CDS searches were performed in only four patients with a DGI, but recommendations were shown for more patients. This is explained by the fact that an alternative drug without a DGI was prescribed following the recommendation shown or because drugs were not prescribed directly following the search. The latter is illustrated by the search types we could distinguish. Some searches concerned past or future treatment, and prescribers also checked drugs they did not want to prescribe at that moment, for example commonly used treatment alternatives or drugs that were suggested in a recommendation. Furthermore, it is likely that prescribers started to remember the recommendations for DGIs they had encountered previously and did not perform a CDS search every time. The number of CDS searches reported is therefore likely to be an underestimation of the actual number of times prescribers dealt with PGx results.  

From the actionable recommendations evaluated, we conclude that DPWG guidelines are generally well adhered to, although practical application can transcend guideline recommendations and application is thus not always straightforward.’ 

Q2 Secondly, it is not easy to find how you hoped to identify new(?) barriers and in which group. Table S2 mentions the questions but I fail to understand what the answers mean (eg fgen_apoth_16)

A2 We agree that Table S2 is confusing, since although it included the question identifiers (e.g. fgen_apoth_16) it did not list the possible multiple-choice answers. We have replaced the identifiers with the (translated) answer options that participants could select. 

Q3 In the results there is a mixture of identified barriers to apply PGx and problems with implementation of the results eg patients attitude towards PGx and need for further information about the results. Last point is very interesting but does it refrain a patient from taking the PGx? 

A3 In our study, the need for further information about the results expressed by patients refers exclusively to the provided explanation of the implications for personal use of medication after receiving their PGx screening results, rather than referring to information provided at the time of participant recruitment. In addition, we did not assess the reason why patients declined participating in this study and therefore cannot speculate on whether any lack of study information details, on PGx and/or study logistics, provided at that point,  played a role in non-participation. 

Q4 A table with abbreviations would be helpful

A4 We acknowledge that some of the uncommon abbreviations may be confusing to readers. To improve readability, we have not abbreviated ‘healthcare practitioner(s)’ in the manuscript. As suggested, we have also added the following table with abbreviations: 

Table S5 Alphabetical list of abbreviations

Abbreviation

Explanation

ADR

Adverse drug reaction

CDS

Clinical Decision Support

DPWG

Dutch Pharmacogenetics Working Group

EHR

Electronic Health Record

GP

General practitioner

ICT

Information and communications technology

PGx

Pharmacogenomics

T1, T2

Timepoint 1, timepoint 2

UMCG

University Medical Center Groningen

In line with this comment, we made changes to the figure and table captions of figure 2, figure 3, table 2 and table 3, as follows: 

Figure 2: Conversation scores for the discussion of pharmacogenomic test results with healthcare practitioners. The number of conversations between patients and different healthcare practitioners, the score patients gave to those conversations, and the supporting arguments for the score given.’ (line 326)

Figure 3: The steps which community pharmacists reported they would take after having received pharmacogenomic screening results.’ (line 370)

Table 2: Final responsibility for the application of pharmacogenomic screening results in patient care.’ (line 382)

Table 3: Barriers and facilitators to pharmacogenomic screening implementation’ (line 401) 

Q5 Supportive material in Dutch is not relevant for most readers.

A5 We agree that the Dutch materials are not relevant for most readers, therefore we have removed these materials. We removed the references to these materials and instead included the following statement in the methods section (line 101-102): 

‘Copies of these materials (in Dutch) are available upon request.’

Reviewer 3 Report

Very nice work about personalized medicine regarding pharmacotherapy in medicine and psychiatry, analysing practical barriers and facilitators, and showing solutions for improved use

Line 140 show low adherence to questionnaires. are these numbers correct? , numbers discussed later in the text  (line 268, for instance, show better adherence)

Supplementary Methods have 2 section 5

Author Response

Very nice work about personalized medicine regarding pharmacotherapy in medicine and psychiatry, analysing practical barriers and facilitators, and showing solutions for improved use 

We thank the reviewer for supporting the importance of the topic of our work and provide a point-by-point response to the comments below. 

Q1 Line 140 show low adherence to questionnaires. are these numbers correct? , numbers discussed later in the text  (line 268, for instance, show better adherence).

A1 The response rates are correct, but we understand the source of the confusion. The percentages in line 162-163 (line 140 in the original submission) reflect the proportion of invited healthcare practitioners that subsequently participated in our study. In contrast, the percentages mentioned throughout the rest of the text do not reflect the proportion of invited healthcare practitioners, but the proportion of responses of the participants to each individual question, as not all responders answered each question.
To avoid any confusion among our readers, we have now explicitly mentioned at the first occurrence in every paragraph of the results section that the percentages reflect the proportion of respondents for the individual questions, e.g. ‘Sixteen community pharmacists (36% of respondents)’ (line 331). Also, we have included which timepoint was used (T1 or T2) for the questions. 

Q2 Supplementary Methods have 2 section 5.

A2 We thank the reviewer for spotting this mistake and have corrected the section headers in the Supplementary Methods as follows:

‘5. Clinical Decision Support system’ and ‘6. Data collection’.

Additional changes to the manuscript

We would like to inform the editor and reviewers that we also applied the following additional changes:

  • We added the following facilitator to Table 3 in line 401, because we also mention this facilitator in the discussion and abstract as one of the main findings: ‘DPWG guidelines are generally well adhered to’.
  • As requested by the editor, we also include a graphical abstract.
  • Due to the removal of figure 2, we have changed the numbering of figure 3 and 4 into figure 2 and 3, respectively. (lines 324, 326, 361 and 370)
  • Reference [17] is no longer mentioned in the text, so it was removed from the references section.

We hope that our response meets your expectation and make our manuscript suitable for publication in the Journal of Personalized Medicine.

Round 2

Reviewer 2 Report

no further suggestions